# $\zeta$-DP: Complex-Valued Differential Privacy and its Applications to Neural Network Training

## Abstract

We present $\zeta$-DP, an extension of differential privacy (DP) to complex-valued functions. After introducing the complex Gaussian mechanism, whose properties we characterise in terms of $(\varepsilon, \delta)$-DP, Rényi DP and Gaussian DP, we present $\zeta$-DP stochastic gradient descent ($\zeta$-DP-SGD), a variant of DP-SGD for training complex-valued neural networks. We experimentally evaluate $\zeta$-DP-SGD on three complex-valued tasks, i.e. electrocardiogram classification, speech classification and magnetic resonance imaging (MRI) reconstruction. Moreover, we benchmark the performance of our methods on real- and complex-valued variants of the CIFAR-10 and MNIST datasets as well as a large range of complex-valued activation functions. Our experiments demonstrate that DP training of complex-valued neural networks is possible with rigorous privacy guarantees and excellent utility and represents a promising technique to mitigate privacy-utility trade-offs.

## 1 Introduction

The ability to harness diverse, feature-rich datasets for algorithm training can allow the scientific community to create machine learning (ML) models capable of solving challenging data-driven tasks. These include the creation of robust autonomous vehicles (Rao & Frtunikj, 2018), early-stage cancer discovery (Cruz & Wishart, 2006) or disease survival prediction (Rau et al., 2018). A subclass of these ML problems is able to profit particularly from the ability to execute deep learning workflows over complex-valued datasets, such as magnetic resonance imaging (MRI) (Virtue et al., 2017) or time-series data (Fan & Xiong, 2013; Kociuba & Rowe, 2016). Complex-valued deep learning has seen increased traction in the past years, owing in part to the improved support by ML frameworks and the broader availability of graphics processing unit (GPU) hardware able to tackle the increased computational requirements (Bassey et al., 2021). However, since complex numbers are often used to represent signals derived from sensitive biological or medical records (Cole et al., 2020; Küstner et al., 2020; Peker, 2016), privacy constraints can render such datasets hard to obtain. The resulting data scarcity impairs effective model training, prompting the adoption of regulation-compliant and privacy-preserving methods for data access.

Distributed computation methods such as federated learning (FL) (Konečný et al., 2016) can partially address this requirement by only requiring participants to share results of the local computation rather than exchange data over the network. However, FL on its own has repeatedly been shown to be insufficient in the task of privacy protection (Geiping et al., 2020; Yin et al., 2021). Thus, bridging the gap between data protection and utilisation for algorithmic training requires methods able to offer objective privacy guarantees. Differential privacy (DP) (Dwork et al., 2014) has established itself as the cornerstone of such techniques and has been deployed in contexts like the US Census (Abowd, 2018) and distributed learning on mobile devices (Cormode et al., 2018). DP's purview has been expanded to encompass deep learning through the introduction of DP stochastic gradient descent (DP-SGD) (Abadi et al., 2016), allowing for the training of deep neural networks on private data. So far, however, the application of DP to complex-valued ML tasks remains drastically under-explored.

Our work attempts to address this challenge through the following contributions:

1. We extend DP to the complex domain through a collection of techniques we refer to as $\zeta$-DP. We use this term instead of "complex-valued DP" for brevity and to avoid confusion with the abbreviation "cDP", which is already used for concentrated DP (Dwork & Rothblum, 2016). The letter $\zeta$ alludes to the complex-valued *Riemann* $\zeta$ function and is intended to convey the notion of "continuation" to the complex domain.

2. We define and discuss the complex Gaussian Mechanism (GM) in Section 4.1 and show that its properties generalise corresponding results on real-valued functions. This allows us to interpret the complex GM through the lens of previous work on $(\varepsilon, \delta)$-DP, Rényi-DP (RDP) and Gaussian DP (GDP).

3. To enable the design and privacy-preserving training of complex-valued deep learning models, we introduce $\zeta$-DP-SGD in Section 4.2.

4. In Section 5 we experimentally evaluate the aforementioned techniques on several real-life neural network training tasks, i.e. speech classification, abnormality detection in electrocardiograms and magnetic resonance imaging (MRI) reconstruction. Moreover, we establish baselines for future work by providing benchmark results on real- and complex-valued variants of the MNIST and CIFAR-10 datasets and on complex neural network activation functions both with and without $\zeta$-DP-SGD.

5. We conclude that our DP analysis and the utilisation of tight sensitivity accounting through the Wirtinger/$\mathbb{CR}$-calculus allows practitioners to avail themselves of the higher learning capacity of neural networks with complex-valued weights to achieve improved utility with the same privacy guarantees as corresponding real-valued architectures.

## 2 RELATED WORK

Prior work has addressed several challenges in non-private complex-valued deep learning, including the introduction of appropriate activation functions, and has presented applications in domains such as MRI reconstruction (Küstner et al., 2020) or time series analysis (Fink et al., 2014). For a detailed overview of methodology and applications, we refer to Hirose (2012); Bassey et al. (2021). Until recently, deep learning frameworks did not fully support complex arithmetic and automatic differentiation. Hence, previous works (Trabelsi et al., 2017; Nazarov & Burnaev, 2020) express $\mathbb{C}^n$ as $\mathbb{R}^{2n}$ and use two real-valued channels rather than complex floating-point numbers. This approach can lead to a spurious increase in function sensitivity by incorrectly computing gradient magnitudes and, by extension, to the addition of excessive noise in the private setting, adversely impacting utility. Our work specifically addresses this shortcoming through the use of complex-valued weights and the Wirtinger/$\mathbb{CR}$-calculus (Wirtinger, 1927; Kreutz-Delgado, 2009). Only a limited number of studies have utilised DP techniques in conjunction with complex-valued data (Fan & Xiong, 2013; Fioretto et al., 2019), however, to our knowledge none has formalised a notion of complex-valued DP or investigated neural network applications.

The $(\varepsilon, \delta)$-definition of DP and the Gaussian mechanism are essential to our formalism, and details on their real-valued definitions can be found in Dwork et al. (2014). As stated above, DP-SGD was introduced by Abadi et al. (2016). Rényi-DP (RDP) was introduced by Mironov (2017) as a relaxation of $(\varepsilon, \delta)$-DP with favourable properties under composition, rendering it particularly useful for DP-SGD privacy accounting. Gaussian DP (GDP) was introduced by Dong et al. (2019) and tailored to the specific properties of the Gaussian mechanism, for which it provides a tight privacy analysis.

## 3 BACKGROUND

We begin by introducing key terminology required in the rest of our work. We assume that a trusted analyst in possession of sensitive data wishes to publish the results of some analysis performed on this data while offering the individuals to whom the data belongs a DP guarantee. We will refer to the set of all sensitive records as the *sensitive database $D$*, whereby we assume that one individual's data is only present in the database once. Let $X$, the metric space of all sensitive databases, be equipped with the Hamming metric $d_X$ and let $D \in X$. $D$'s *adjacent* database $D'$ can be constructed from $D$ by adding or removing exactly one database row (that is, one individual's data), such that

$d_X(D, D') = 1$. The analyst executes a *query (function)* $f$, for example a mean calculation, over the database. We first define the *sensitivity* of $f$:

**Definition 1** (Sensitivity $\Delta$ of $f$). *Let $X$ and $d_X$ be defined as above. $f$ maps the elements of $X$ to elements of a metric space $Y$ equipped with a metric $d_Y$. The (global) sensitivity $\Delta$ of $f$ is then defined as:*

$$\Delta(f) = \max_{D, D' \in X} \frac{d_Y(f(D), f(D'))}{d_X(D, D')}, D \neq D'. \tag{1}$$

*The maximum is taken over all adjacent database pairs in $X$. When $Y$ is the Euclidean space and $d_Y$ is the $L_2$ metric, $\Delta$ is referred to as the $L_2$ sensitivity. We will only use the $L_2$ sensitivity in this work.*

In private data analysis and ML, we are often concerned with differentiable functions; for Lipschitz-continuous query functions, the equivalence of the Lipschitz constant and the $L_2$-sensitivity (Raskhodnikova & Smith, 2016) can be exploited:

**Definition 2** (Lipschitz constant $K$ of $f$). *Let $X, Y, d_X$ and $d_Y$ be defined as above. Then $f$ is said to be $K$-Lipschitz continuous if and only if a non-negative real number $K_f$ exists for which the following holds:*

$$d_Y(f(D), f(D')) \leqslant K_f \, d_X(D, D'). \tag{2}$$

Evidently, $K_f \equiv \Delta$ by Equation (1) and the definition of adjacency. Moreover, let $\mathcal{D}$ be the differential operator; then $K_f = \sup \|\mathcal{D}(f)\|$, where $\| \cdot \|$ is the operator norm (O'Searcoid, 2006). Therefore, for a scalar-valued query function, $\Delta \equiv K_f \equiv \sup \|\nabla f\|_2$.

A DP *mechanism* adds noise to the results of $f$ calibrated to its sensitivity. Here, we provide the definition of the (real-valued) Gaussian mechanism (GM):

**Definition 3** (Gaussian mechanism). *The Gaussian mechanism $\mathcal{M}$ operates on the results of a query function $f : \mathbb{R}^n \mapsto \mathbb{R}^d$ with sensitivity $\Delta$ over a sensitive database $D$ by outputting $f(D) + \xi$, where $\xi \sim \mathcal{N}(0, \sigma^2 \, \mathbf{I}_d)$. Here, $\mathbf{I}_d$ denotes the identity matrix with $d$ diagonal elements and $\sigma^2$ is the variance of Gaussian noise calibrated to $\Delta$.*

The application of the GM with properly calibrated noise satisfies $(\varepsilon, \delta)$-DP:

**Definition 4** ($(\varepsilon, \delta)$-DP). *The randomised mechanism $\mathcal{M}$ preserves $(\epsilon, \delta)$-DP if, for all pairs of inputs $D$ and $D'$ and all subsets $\mathcal{S}$ of $\mathcal{M}$'s range:*

$$\mathbb{P}(\mathcal{M}(f(D) \in \mathcal{S})) \leqslant e^{\varepsilon} \, \mathbb{P}(\mathcal{M}(f(D') \in \mathcal{S})) + \delta. \tag{3}$$

A number of *relaxations* have been proposed to characterise the properties of the GM, of which *Rényi DP* is arguably the most widely employed in DP deep learning frameworks owing to its favourable properties under composition.

**Definition 5** (Rényi DP). *$\mathcal{M}$ preserves $(\alpha, \rho)$-Rényi-DP (RDP) if, for all pairs of inputs $D$ and $D'$:*

$$D_\alpha \left( \mathcal{M}(f(D)) \, \| \, \mathcal{M}(f(D')) \right) \leqslant \rho \tag{4}$$

*where $D_\alpha$ denotes the Rényi divergence of order $\alpha > 1$ between $\mathcal{M}(f(D))$ and $\mathcal{M}(f(D'))$. At $\alpha = 1$, the divergence is defined by continuity as the Kullback-Leibler divergence. We note that the term "between" is an abuse of terminology, as the Rényi divergence is asymmetric. In general, we will use $D_\alpha(p, q)$ to denote $\sup\{D_\alpha(p, q), D_\alpha(q, p)\}$.*

Gaussian DP (GDP) was introduced as a variant of f-DP by Dong et al. (2019) specifically tailored to the properties of the GM, and provides the tightest possible characterisation of its properties. Relying on statistical hypothesis testing, f-DP interprets DP through a *trade-off function* between the *Type I* and *Type II* statistical errors faced by an adversary trying to determine whether one of the adjacent databases contains the individual or not. GDP is a specialisation of f-DP when the trade-off-function has the form $G_\mu := T(\mathcal{N}(0, 1), \mathcal{N}(\mu, 1))$.

**Definition 6** (Gaussian DP). *$\mathcal{M}$ preserves $\mu$-Gaussian DP (GDP) if, for all pairs of adjacent databases $D$ and $D'$:*

$$T(\mathcal{M}(f(D)), \mathcal{M}(f(D'))) \geqslant G_\mu \tag{5}$$

*where $T$ denotes a trade-off function. In this case, $G_\mu(\alpha) = \Phi \left( \Phi^{-1}(1 - \alpha) - \mu \right)$ where $\alpha$ is the Type I statistical error and $\Phi$ is the cumulative distribution of the standard, real-valued normal distribution.*

## 4 $\zeta$-DP

In this section we introduce $\zeta$-DP, an extension of DP to complex-valued query functions and mechanisms. $\zeta$-DP generalises real-valued DP and allows the re-use of prior theoretical results and software implementations.

### 4.1 THE COMPLEX GAUSSIAN MECHANISM

We begin by introducing a variant of the GM suitable to query functions with codomain $\mathbb{C}$.

**Definition 7** (Complex Gaussian mechanism). *The complex Gaussian mechanism $\mathcal{M}_{\mathbb{C}}$ on $f : \mathbb{C}^n \mapsto \mathbb{C}^d$ outputs $f(D)+\xi$, where $\xi \sim \mathcal{N}_{\mathbb{C}}(0, \sigma^2 \, \mathbf{I}_d)$ and $\mathcal{N}_{\mathbb{C}}(0, \sigma^2)$ denotes circularly symmetric complex-valued Gaussian noise with variance $\sigma^2$.*

Of note, a random variable $X \sim \mathcal{N}_{\mathbb{C}}(0, \sigma^2)$ can be constructed by independently drawing two random variables $A$, $B$ from a real-valued normal distribution $\mathcal{N}(0, \frac{\sigma^2}{2})$ and outputting $X = A+B\,\mathbf{i}$, where $\mathbf{i}$ is the imaginary unit. This property is unique to the complex GM and leverages the fact that the inner product in $\mathbb{C}^n$ is non-bilinear to add noise scaled by $\sqrt{2} \cdot \sigma$ to each component of the complex number. Naively "simulating" $\mathbb{C}^n$ as $\mathbb{R}^{2n}$, the latter being equipped with a bi-linear inner product, would instead require the addition of independent Gaussian noise to each component of a vector with resulting scale $2 \cdot \sigma$.

We now state our main theoretical results and proof sketches. Detailed proofs can be found in Appendix A.1.

**Theorem 1.** *Let $f : \mathbb{C}^n \mapsto \mathbb{C}^d$ be a query function with sensitivity $\Delta$. Then, $\mathcal{M}_{\mathbb{C}}$ preserves $(\varepsilon, \delta(\varepsilon))$-DP if and only if the following holds $\forall \, \varepsilon > 0, \delta \in [0, 1]$:*

$$\delta(\varepsilon) \geqslant \Phi\left(\frac{\Delta}{2\sigma} - \frac{\varepsilon\sigma}{\Delta}\right) - e^\varepsilon \Phi\left(-\frac{\Delta}{2\sigma} - \frac{\varepsilon\sigma}{\Delta}\right) \tag{6}$$

*where $\Phi$ denotes the cumulative distribution function of the standard (real-valued) normal distribution.*

*Proof (Sketch).* It suffices to show that the magnitude of the privacy-loss random variable $\Omega$ is bounded by $\varepsilon$ with probability $1 - \delta$. This holds, as $\Omega$ is distributed as a real-valued normal distribution even when $O \in \mathbb{C}$, where $O$ is any output of $f$, seeing as the mean of $\Omega$ is distributed as $(f(D) - f(D'))(f(D) - f(D'))^H \in \mathbb{R}$, where $H$ denotes the Hermitian transpose. $\square$

**Theorem 2.** *Let $f$ be defined as above. Then, $\mathcal{M}_{\mathbb{C}}$ preserves $(\alpha, \rho)$-RDP if:*

$$\sigma^2 \geqslant \frac{1}{2}\frac{\alpha\Delta^2}{\rho}. \tag{7}$$

*Proof (Sketch).* The Rényi divergence of order $\alpha > 1$ between two circularly symmetric, complex-valued normal distributions with means $\boldsymbol{\mu}_0 \in \mathbb{C}^n$ and $\boldsymbol{\mu}_1 \in \mathbb{C}^n$ and common variance $\sigma^2\mathbf{I}_n$ is:

$$D_\alpha\left(\mathcal{N}(\boldsymbol{\mu}_0, \sigma^2\mathbf{I}_n) \, \| \, \mathcal{N}(\boldsymbol{\mu}_1, \sigma^2\mathbf{I}_n)\right) = \frac{\alpha\langle\boldsymbol{\mu}_0, \boldsymbol{\mu}_1\rangle}{2\sigma^2} = \tag{8}$$

$$= \frac{\alpha(f(D) - f(D'))(f(D) - f(D'))^H}{2\sigma^2} = \frac{\alpha\|f(D) - f(D')\|_2^2}{2\sigma^2} \leqslant \frac{\alpha\Delta^2}{2\sigma^2}, \tag{9}$$

where $\langle\cdot\rangle$ denotes the inner product. $\square$

**Theorem 3.** *Let $f$ be defined as above, $\Delta$ be the $L_2$-sensitivity of $f$ and $\sigma$ be the standard deviation of $\mathcal{M}_{\mathbb{C}}$'s noise. Let $G_\mu$ be the trade-off function of a $\mu$-GDP real-valued GM. Then, if $\lambda = \Delta/\sigma$, $\mathcal{M}_{\mathbb{C}}$ preserves $\mu$-GDP if and only if $\lambda \leqslant \mu$. Thus, to preserve $\mu$-GDP, one must choose $\sigma^2$ such that:*

$$\sigma^2 \geqslant \left(\frac{\Delta}{\mu}\right)^2. \tag{10}$$

*Proof (Sketch).* We will show that $T\left(\mathcal{N}_{\mathbb{C}}(f(D),\sigma),\mathcal{N}_{\mathbb{C}}(f(D'),\sigma)\right) \geqslant G_\mu$.

$$T\left(\mathcal{N}_{\mathbb{C}}(f(D),\sigma),\mathcal{N}_{\mathbb{C}}(f(D'),\sigma)\right) = G_{\|f(D)-f(D')\|_2/\sigma} \geqslant \tag{11}$$

$$\geqslant G_{\Delta/\sigma} \geqslant G_\mu \Leftrightarrow \frac{\Delta}{\sigma} \leqslant \mu \Rightarrow \lambda \leqslant \mu, \tag{12}$$

which follows from the fact that $\|f(D) - f(D')\|_2 \leqslant \Delta$ and that $G_\mu$ is strictly monotonically decreasing in $\mu$. $\qquad\square$

These findings allow for a seamless transfer of results which apply to real-valued functions to the complex domain. In particular, they yield the following insights:

1. The complex GM inherits all properties of the real-valued GM, such as composition and sub-sampling amplification, as well as the tight analysis afforded by GDP.

2. The complex GM, like the real-valued GM, is fully characterised by the sensitivity $\Delta$ and the magnitude of the noise $\sigma$.

3. The GM "naturally fits" $\zeta$-DP due to the aforementioned convenient properties of the circularly symmetric complex-valued Gaussian distribution. As an additional counterexample, a complex-valued *Laplace* random variable is naturally non-circular in the complex (and multivariate) case, even when constructed from independent distributions (Kotz et al., 2001). Moreover, the utilisation of the $L_1$-metric on the output space of $f$ is disadvantageous, as even for scalar (complex) outputs, the $L_1$ sensitivity can be higher than the $L_2$ sensitivity. Lastly, the utilisation of elliptical Laplace noise is inherently unable to satisfy $(\varepsilon, 0)$-DP in any dimension $> 1$ (Reimherr & Awan, 2019). We thus leave the introduction of alternative strategies for obtaining $(\varepsilon, 0)$-DP in the complex-valued setting to future investigation.

We conclude this section by introducing a modification of the DP stochastic gradient descent (DP-SGD) algorithm, which will be employed in our experimental evaluation.

## 4.2 $\zeta$-DP-SGD

The DP-SGD algorithm (Abadi et al., 2016) represents an application of the GM to the training of deep neural networks. Using the terminology above, each training step of the neural network (which, in this setting, represents the *query*) leads to the release of a privatised gradient. Evidently, the noise magnitude of the GM must be calibrated to the sensitivity of the loss function. However, most neural network loss functions have a Lipschitz constant which is too high to preserve DP while maintaining acceptable utility (and –generally –the Lipschitz constant of neural networks is NP-hard to compute (Scaman & Virmaux, 2018)). Thus, DP-SGD (Abadi et al., 2016) artificially induces a bounded sensitivity condition by *clipping* the $L_2$-norm of the gradient to a pre-defined value. A loss function with real-valued outputs is required for minimisation (even when the function's arguments are complex-valued) as the complex plane –contrary to the real number line– does not admit a natural ordering.

Our implementation of the algorithm makes use of Wirtinger (or $\mathbb{CR}$-) calculus (Kreutz-Delgado, 2009) for gradient computations similar to previous works on complex-valued deep learning (Virtue et al., 2017; Boeddeker et al., 2017). This technique, discussed in detail Appendix A.5, provides several benefits: It relaxes the requirement for component functions to be *holomorphic* (that is, differentiable *in the complex sense*), only requiring them to be individually differentiable with respect to their real and imaginary components (differentiable *in the real sense*). For holomorphic functions $\mathbb{C} \mapsto \mathbb{C}$, $\mathbb{CR}$-derivatives nevertheless recover the correct derivative definition. Thus, $\mathbb{CR}$-derivatives can also be used to compute the global sensitivity in the $\mathbb{C} \mapsto \mathbb{C}$-case via the Lipschitz constant. More importantly, for functions $\mathbb{C} \mapsto \mathbb{R}$, they lead to a correct gradient magnitude calculation, whereas expressing complex-valued functions as vector-valued functions in $\mathbb{R}^{2n}$, a technique often employed in complex-valued neural network training (Trabelsi et al., 2017), can incur an undesirable multiplicative sensitivity increase which would diminish the utility of $\zeta$-DP-SGD. We exemplify this phenomenon and the noise savings $\mathbb{CR}$-calculus can enable in Appendix A.5 and Section 5.1.

$\zeta$-DP-SGD is presented in Algorithm 1 and relies on a modification of the gradient clipping step: we clip the *conjugate gradient*, which represents the direction of steepest ascent for a loss function

$\mathcal{L}$ with real-valued outputs and complex-valued arguments:

$$\nabla \overline{\mathcal{L}} := \left( \frac{\partial \mathcal{L}}{\partial \overline{\theta}_1}, \ldots, \frac{\partial \mathcal{L}}{\partial \overline{\theta}_n} \right) \tag{13}$$

where $\left(\overline{\theta}_1, \ldots, \overline{\theta}_n\right)$ is the conjugate weight vector. We remark that, due to the aforementioned properties of the complex GM, the algorithm is compatible with other first-order optimisers (such as Adam), as well as other clipping techniques, such as per-layer clipping (McMahan et al., 2018). Moreover, newer methods for analysing mechanism composition, such as the *Fourier Accountant* (Koskela et al., 2020) can be used.

---

**Algorithm 1** $\zeta$-DP-SGD

---

**Require:** Database with samples $\{x_1, \ldots, x_N\} \in \mathbb{C}^n$, neural network with loss function $\mathcal{L}$ and weight vector $\boldsymbol{\theta} \in \mathbb{C}^n$. Hyperparameters: learning rate $\eta_t$, noise variance $\sigma^2$, sampling probability $R$, gradient norm bound $B$, total steps $T$.

**Initialize** $\boldsymbol{\theta}_0$ randomly

**for** $t \in [T]$ **do**

    Draw a *lot* $L_t$ with sampling probability $R$ using *Poisson* or uniform sampling

    **Compute per-sample conjugate gradient**

    For each $i \in L_t$, compute $\overline{\boldsymbol{g}}_t(x_i) \leftarrow \nabla \overline{\mathcal{L}}(\boldsymbol{\theta}_t, x_i)$

    **Clip conjugate gradient**

    $\breve{\boldsymbol{g}}_t(x_i) \leftarrow \overline{\boldsymbol{g}}_t(x_i) / \max \left( 1, \frac{\|\overline{\boldsymbol{g}}_t(x_i)\|_2}{B} \right)$

    **Apply the Complex Gaussian Mechanism and average**

    $\widetilde{\boldsymbol{g}}_t \leftarrow \frac{1}{L} \left( \sum_i \breve{\boldsymbol{g}}_t(x_i) + \mathcal{N}_{\mathbb{C}}(0, \sigma^2 B^2 \, \mathbf{I}_L) \right)$

    **Descend**

    $\boldsymbol{\theta}_{t+1} \leftarrow \boldsymbol{\theta}_t - \eta_t \widetilde{\boldsymbol{g}}_t$

**end for**

**Output** updated neural network weight vector $\boldsymbol{\theta}_T$ and compute the privacy cost.

---

## 5 EXPERIMENTAL EVALUATION

Throughout this section, we present results from the experimental evaluation of $\zeta$-DP-SGD. Details on dataset preparation and training can be found in Appendix A.6. All $\varepsilon$-values are converted (losslessly) from GDP. RDP guarantees are slightly looser and provided in Appendix A.2. We also provide an experimental evaluation of commonly used complex-valued activation functions for neural network training with $\zeta$-DP-SGD in Appendix A.3. Lastly, we provide additional benchmark experiments demonstrating a complex-valued variant of the MNIST dataset in Appendix A.4.

### 5.1 BENCHMARKING $\zeta$-DP-SGD ON CIFAR-10

We begin by demonstrating that the utilisation of the complex Gaussian mechanism and $\zeta$-DP-SGD provides increased model utility for the same privacy budget, even on tasks with real-valued inputs in which no additional information from the input's imaginary component is available. Moreover, we empirically confirm that, for $\zeta$-DP-SGD, leveraging the Wirtinger/$\mathbb{CR}$-calculus for gradient computations with respect to complex-valued weights results in a lower sensitivity penalty, and thus improved performance compared to "simulating" complex-valued neural networks with real-valued weights in $\mathbb{R}^{2n}$ similar to Trabelsi et al. (2017). Further theoretical and experimental evaluation of this phenomenon can be found in Appendix A.5.

We selected the CIFAR-10 dataset (Krizhevsky et al., 2009), a challenging dataset in the (real-valued) DP setting. We selected the model architecture recently reported by Papernot et al. (2020), which we equipped with either complex-valued weights (for training in $\mathbb{C}$ or with an additional weight matrix (for training in $\mathbb{R}^{2n}$). We trained to the same $\varepsilon$ values as Papernot et al. (2020) (in RDP), but modified each image such that the real component contained the image and the imaginary component was zero-filled (for training in $\mathbb{C}$), or added an additional trailing zero-filled channel (for training in $\mathbb{R}^{2n}$). The results were identical when the imaginary/second channel components were filled with the same real-valued image. Table 1 summarises these results. Without DP, the increased

learning capacity of the complex-valued weights led to improved test set performance compared to real-valued weights, with identical performance for training in $\mathbb{C}$ and in $\mathbb{R}^{2n}$. When DP was employed, the model trained with $\zeta$-DP-SGD achieved the best performance, outperforming both the real-valued DP model and the $\mathbb{R}^{2n}$ "simulated" complex-valued DP model. This result underscores the aforementioned theoretical finding that the complex Gaussian mechanism (contrary to the naive application of a real-valued multivariate Gaussian mechanism in $\mathbb{R}^{2n}$) provides improved performance for the same privacy budget even in tasks with real-valued input data.

Table 1: Classification results for Sections 5.1, 5.2 and 5.3. $\mathbb{C}$: Complex-valued weights. DP: ($\zeta$-)DP-SGD. For the CIFAR-10 dataset, $\mathbb{R}^{2n}$ indicates a network architecture utilising two channels to simulate complex inputs/ model weights. GDP: Gaussian DP accounting. ✗: not used. ✓: used.

| Dataset | $\mathbb{C}$ | DP | Accuracy | ROC-AUC | $F_1$-Score | Recall | GDP $\varepsilon$ | $\delta$ |
|---|---|---|---|---|---|---|---|---|
| | ✗ | ✗ | 80% | 97% | 80% | 80% | $\infty$ | 0 |
| | ✓ | ✗ | **82%** | **98%** | **82%** | **82%** | $\infty$ | 0 |
| CIFAR-10 | ✗ | ✓ | 58% | 91% | 58% | 59% | 7.54 | $10^{-5}$ |
| | ✓ | ✓ | **61%** | **92%** | **61%** | **62%** | 7.54 | $10^{-5}$ |
| | $\mathbb{R}^{2n}$ | ✓ | 58% | 90% | 58% | 57% | 7.54 | $10^{-5}$ |
| ECG | ✓ | ✗ | 87.6% | 91.9% | 70.8% | 81.0% | $\infty$ | 0 |
| | ✓ | ✓ | 88.5% | 92.1% | 69.8% | 71.4% | 1.62 | $5 \cdot 10^{-4}$ |
| SpeechCommands | ✓ | ✗ | 83.4% | 98.0% | 83.2% | 83.3% | $\infty$ | 0 |
| | ✓ | ✓ | 62.5% | 93.7% | 60.3% | 62.6% | 1.39 | $10^{-5}$ |

## 5.2 PRIVACY-PRESERVING ELECTROCARDIOGRAM ABNORMALITY DETECTION ON WEARABLE DEVICES

The advent of wearable devices incorporating electrocardiography (ECG) sensors has provided consumers the ability to detect signs of an abnormal heart rhythm. In this section, we demonstrate the utilisation of a small neural network architecture suitable for deployment, e.g. to a mobile device connected to such a biosensor, to be trained on ECG data from the the China Physiological Signal Challenge (CPSC) 2018 challenge dataset (Liu et al., 2018). We selected the task of automated Left Bundle Branch Block (LBBB) detection, formulated as a binary classification task against a normal (sinus) rhythm. This task is clinically relevant, as the sudden appearance of LBBB can herald acute coronary syndrome which requires urgent attention to avert myocardial infarction. As ECG data constitutes personal health information, its protection is mandated both legally and ethically. We utilised $\zeta$-DP-SGD for training a complex-valued neural network on Fourier-transformed ECG acquisitions. We adopt this strategy as it can benefit from two key properties of the Fourier transform: ECG data can contain high-frequency noise which is irrelevant for diagnosis and can be reduced using Fourier filtering. Concurrently, this technique compresses the signal, which can drastically reduce the amount of data transferred. Table 1 shows classification results and Figure 1 shows exemplary source data.

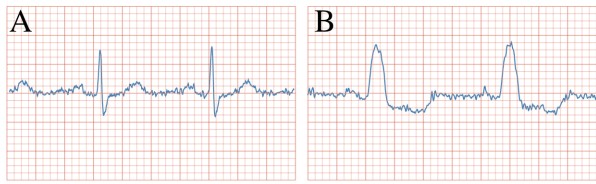

Figure 1: Exemplary ECG data used for classification. **A** shows an example of a sinus rhythm (normal ECG) and **B** shows an example of an ECG exhibiting signs of LBBB. Observe also the high frequency noise around the baseline which can be filtered using the Fourier transform.

### 5.3 DIFFERENTIALLY PRIVATE SPEECH COMMAND CLASSIFICATION FOR VOICE ASSISTANT APPLICATIONS

In recent years, voice assistants have gained popularity in consumer applications such as home speakers, and rely heavily on ML. Recordings collected from users for training speech processing algorithms can be used in impersonation attacks, resulting in successful identity theft (Sweet, 2016) or in *acoustic attacks*, which trigger unintended behaviour in voice assistants (Yuan et al., 2018; Carlini et al., 2016). Protecting privacy in this setting is therefore paramount to increase trust and applicability, as well as safeguard both users and systems from adversarial interference. Convolutional neural networks (CNNs) have been demonstrated to yield state-of-the-art performance on spectrogram-transformed audio data (Palanisamy et al., 2020). However, this and other works (Zhou et al., 2021) typically discard the imaginary components. We here experimentally demonstrate the DP training of a 2-dimensional CNN directly on the complex spectrogram data. We utilised a subset of the SpeechCommands dataset (Warden, 2018), specifically samples from the categories "Yes", "No", "Up", "Down", "Left", "Right", "On", "Off", "Stop", and "Go", summing up to 8000 examples. We transformed each waveform signal to a complex-valued 2-D spectrogram and used $\zeta$-DP-SGD to train a complex-valued CNN. These results are summarised in Table 1 and Figure 2.

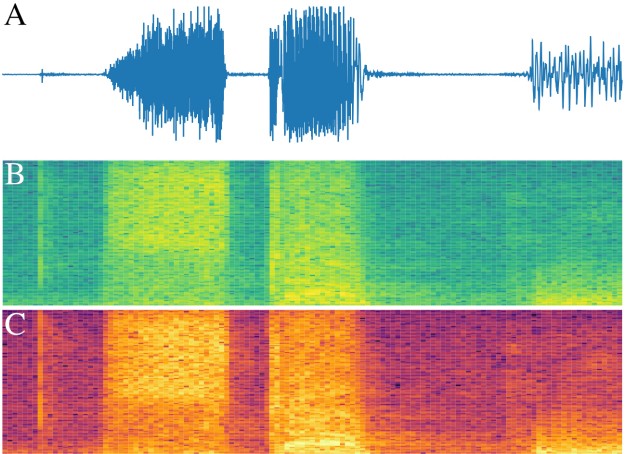

Figure 2: Exemplary waveform (**A**), real (**B**) and imaginary (**C**) spectrogram components from the utterance *Stop*. Spectrograms are log-magnitude transformed for clarity.

### 5.4 MRI RECONSTRUCTION

MRI is an important medical imaging modality and has been studied extensively in the context of deep learning (Akçakaya et al., 2019; Hammernik et al., 2018; Küstner et al., 2020; Muckley et al., 2020). MRI data is acquired in the so-called *k*-space. Sampling only a subset of *k*-space data allows for a considerable speed-up in acquisition time, benefiting patient comfort and costs, however, typically leads to image artifacts, which reduce the diagnostic quality of the resulting MR images. Although neural networks have the ability to produce high-quality reconstructions, their usage for this task has been shown to sometimes lead to the appearance of spurious image content from the fully-sampled reference images the models have been originally trained on (Hammernik et al., 2021; Muckley et al., 2020; Shimron et al., 2021). DP could counteract such *hallucination* as it is designed to limit the effect of individual training examples on model training. However, this positive effect of DP may be counterbalanced by an unacceptable decrease in the diagnostic suitability of the reconstructed images. In this section, we investigate the ramifications of DP on the quality of MRI reconstructions. For this purpose, we trained a complex-valued *U-Net* model architecture on the task of reconstructing single-coil knee MRI images from the *fastMRI* dataset (Zbontar et al., 2018) using pseudo-random *k*-space sampling at $4\times$ acceleration. We observed a nearly equivalent performance in the non-DP and the $\zeta$-DP-SGD settings, whereby the non-DP model enjoyed a $< 2\%$ performance advantage in all metrics. Moreover, to assess the diagnostic suitability of the reconstructed images, we asked a diagnostic radiologist who was blinded to whether or not $\zeta$-DP-SGD was used,

to compare the resulting scans. No differences in diagnostic suitability were observed by the expert in any of the reconstructed images.

We thus conclude that –at least with respect to image quality– DP can indeed match the non-private training of MRI reconstruction models, even at $\varepsilon \ll 1$; we intend to investigate its effect on preventing training data *hallucination* into reconstructed images in future work. Results from these experiments are summarised in Table 2 and Figure 3.

Table 2: Results on the MRI reconstruction task. NMSE: normalised mean squared error, PSNR: peak signal-to-noise ratio in dB, SSIM: structural similarity index metric. GDP: Gaussian DP.

|  | NMSE | PSNR | SSIM | GDP $\varepsilon$ | $\delta$ |
|---|---|---|---|---|---|
| Non-DP | 0.042 | 30.74 | 0.70 | $\infty$ | 0 |
| $\zeta$-DP-SGD | 0.043 | 30.57 | 0.69 | 0.16 | $10^{-5}$ |

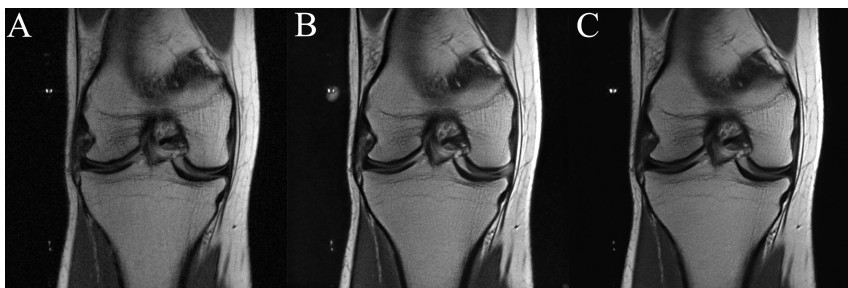

Figure 3: Exemplary reconstruction from a coronal proton-density weighted image of the knee. **A:** reference image, **B:** reconstruction model trained non-privately, **C:** reconstruction model trained with $\zeta$-DP-SGD reconstruction.

## 6 CONCLUSION

Our work presents $\zeta$-DP, an extension of DP to the complex domain and introduces key *building blocks* of DP model training, namely the complex Gaussian mechanism and $\zeta$-DP-SGD. Our experiments demonstrate that the training of DP complex-valued neural networks is possible with high utility under tight privacy guarantees. Our theoretical analysis allows us to leverage the increased learning capacity of complex-weighted neural networks. We thus show improved learning performance compared to scalar real-valued networks and to networks using two real-valued channels to approximate complex numbers, hence, ignoring the specific relationship between the real and imaginary complex components. Lastly, Wirtinger calculus and Gaussian DP allow us to derive the tight sensitivity estimates and privacy bounds for the complex Gaussian mechanism.

We acknowledge some limitations of our work: Both complex-valued deep learning and DP incur a considerable computational performance penalty. Despite steadily improving complex number support, current deep learning frameworks have not yet implemented a full palette of complex-valued layers and activation functions. Moreover, the software framework utilised to computationally realise $\zeta$-DP-SGD in our work relies on multithreading, which suffers from considerable overhead compared to implementations utilising vector instructions and/or bespoke hardware. We discuss the topic of software implementation and provide computational performance benchmarks in Appendices A.7 and A.8. Our investigation highlights a requirement for mature software frameworks able to offer feature and performance parity with their real-valued counterparts, which we intend to develop and publish as free-and-open-source software in the near future.

In conclusion, we contend that $\zeta$-DP represents a promising future research direction for complex-valued and even for purely real-valued tasks, and the improved privacy-utility trade-offs resulting from our work represent a worthwhile contribution to the implementation of DP to a broad variety of relevant tasks, which can help to increase the amount of data available for scientific studies.

## ETHICS STATEMENT

Our work follows all applicable ethical research standards and laws. All experiments were conducted on publicly available datasets. No new data concerning human or animal subjects was generated during our investigation.

## REPRODUCIBILITY STATEMENT

We adhere to ICLRs reproducibility standards and include all necessary information to reproduce our experimental and theoretical results either in the main manuscript or in the Appendix. Theoretical results and proofs can be found in the main manuscript, Section 4 and additional information can be found in Appendix A.5. Details of dataset preparation and analysis can be found in Appendix A.6. Specifically, it contains details about the used datasets, their number of samples, all training, validation and test splits, as well as preprocessing steps. Furthermore, we describe model architectures, employed optimisers, learning rates, and the number of epochs for which models were trained. Lastly, for all DP trainings we provide the noise multipliers, $L_2$ clipping norms and sampling rates, as well as the $\delta$-values at which the $\varepsilon$-values were calculated. Software implementation details and computational resources used can be found in Appendices A.7 and A.8.

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

## A  APPENDIX

### A.1  PROOFS OF THEORETICAL RESULTS

*Proof of Theorem 1.* The claim represents a generalisation of the *Analytic Gaussian Mechanism* (Balle & Wang, 2018) to $\mathcal{M}_\mathbb{C}$. It suffices to show that the magnitude of the *privacy-loss random variable* $\Omega$ is bounded by $\varepsilon$ with probability $1 - \delta$. As shown in Dwork & Rothblum (2016) and Balle & Wang (2018), given some fixed output $O \in \mathbb{C}^d$, $\Omega$ is given by:

$$\Omega = \log \left( \frac{\mathbb{P}(\mathcal{M}_\mathbb{C}(f(D)) = O)}{\mathbb{P}(\mathcal{M}_\mathbb{C}(f(D')) = O)} \right) \tag{14}$$

where $\log$ is the natural logarithm, and is distributed as:

$$\mathcal{N} \left( \frac{\|f(D) - f(D')\|_2^2}{2\sigma^2}, \frac{\|f(D) - f(D')\|_2^2}{\sigma^2} \right). \tag{15}$$

As $\|f(D) - f(D')\|_2^2 = (f(D) - f(D'))(f(D) - f(D'))^H \in \mathbb{R}$, where $H$ denotes the Hermitian transpose, $\Omega$ has a real-valued mean and hence follows a real-valued normal distribution, even when $O \in \mathbb{C}$. From here, the proof proceeds identically to the proof to Theorem 8 of (Balle & Wang, 2018). $\square$

For proving Theorem 2, we will rely on the following fact about the Rényi divergence of order $\alpha$ between arbitrary distributions:

**Corollary 1** (Definition 2 in (Van Erven & Harremos, 2014)). *Let $P$ and $Q$ be two arbitrary distributions defined on a measurable space $(\mathcal{X}, \mathcal{F})$ with densities $p(x)$ and $q(x)$. Then, for $\alpha > 1$:*

$$D_\alpha(P \parallel Q) = \frac{1}{\alpha - 1} \log \int_{-\infty}^\infty p(x)^\alpha q(x)^{1-\alpha} dx. \tag{16}$$

*In particular, for two normal distributions with means $\boldsymbol{\mu}_0$ and $\boldsymbol{\mu}_1$ and common variance $\sigma^2 \mathbf{I}$:*

$$D_\alpha \left( \mathcal{N}(\boldsymbol{\mu}_0, \sigma^2 \mathbf{I}) \parallel \mathcal{N}(\boldsymbol{\mu}_1, \sigma^2 \mathbf{I}) \right) = \frac{\alpha \langle \boldsymbol{\mu}_0, \boldsymbol{\mu}_1 \rangle}{2\sigma^2} \tag{17}$$

*where $\langle \cdot \rangle$ denotes the inner product.*

We can now prove Theorem 2:

*Proof of Theorem 2.* By Definition 7 and the additive property of the Gaussian distribution, the density functions of $\mathcal{M}_\mathbb{C}$ on $f(D)$ and $f(D')$ follow a circularly symmetric complex-valued Gaussian distribution with means $f(D)$ and $f(D')$ and common covariance matrix $\sigma^2 \mathbf{I}$. By substituting in equation (17):

$$D_\alpha \left( \mathcal{N}_\mathbb{C}(f(D), \sigma^2) \parallel \mathcal{N}_\mathbb{C}(f(D'), \sigma^2) \right) = \frac{\alpha \langle f(D), f(D') \rangle}{2\sigma^2} =$$
$$= \frac{\alpha(f(D) - f(D'))(f(D) - f(D'))^H}{2\sigma^2} = \frac{\alpha \|f(D) - f(D')\|_2^2}{2\sigma^2} \leqslant \frac{\alpha \Delta^2}{2\sigma^2}. \tag{18}$$

Hence, to preserve $(\alpha, \rho)$-RDP, it suffices to choose $\sigma^2$ such that $\sigma^2 \geqslant \alpha \Delta^2 / 2\rho$. $\square$

*Proof of Theorem 3.* We recall that the trade-off function between $\mathcal{M}_\mathbb{C}(f(D))$ and $\mathcal{M}_\mathbb{C}(f(D'))$ is given by:

$$G_\mu := T \left( \mathcal{N}(0, 1), \mathcal{N}(\mu, 1) \right), \tag{19}$$

and is strictly monotonically decreasing in $\mu$ such that if $\mu_1 \geqslant \mu_2$, then $G_{\mu_1} \leqslant G_{\mu_2}$. To preserve $\mu$-GDP, we require that:

$$T \left( \mathcal{N}_\mathbb{C}(f(D), \sigma), \mathcal{N}_\mathbb{C}(f(D'), \sigma) \right) \geqslant G_\mu, \tag{20}$$

intuitively, that the outputs of $\mathcal{M}_{\mathbb{C}}$ are *at least as hard* to distinguish as the distributions $\mathcal{N}(0,1)$ and $\mathcal{N}(\mu,1)$ given a single draw. We have:

$$T\left(\mathcal{N}_{\mathbb{C}}(f(D),\sigma),\mathcal{N}_{\mathbb{C}}(f(D'),\sigma)\right) = \Phi\left(\Phi^{-1}(1-\alpha) - \frac{\|f(D)-f(D')\|_2}{\sigma})\right) \quad (21)$$

$$= G_{\|f(D)-f(D')\|_2/\sigma}. \quad (22)$$

We require

$$G_{\|f(D)-f(D')\|_2/\sigma} \geqslant G_{\Delta/\sigma} \geqslant G_\mu \Leftrightarrow \frac{\Delta}{\sigma} \leqslant \mu \Rightarrow \lambda \leqslant \mu, \quad (23)$$

where we have used the fact that $\|f(D)-f(D')\|_2 \leqslant \Delta$ and the strict monotonicity of $G_\mu$. Thus, to preserve $\mu$-GDP, one must choose $\sigma^2$ such that $\sigma^2 \geqslant (\Delta/\mu)^2$. $\qquad\square$

## A.2 RÉNYI DP ANALYSIS OF THE MAIN MANUSCRIPT EXPERIMENTS

Table 3 provides an overview of the differences between the privacy bounds of RDP and GDP for the experiments in the main manuscript. We recall that GDP provides a tight analysis of the Gaussian mechanism (Dong et al., 2019) and a lossless conversion between GDP and $(\varepsilon,\delta)$-DP is possible, whereas RDP cannot be losslesly converted. Of note, we are already using the improved conversion formula from RDP to $(\varepsilon,\delta)$-DP shown in Balle et al. (2020) instead of the original formula proposed in Mironov (2017).

Table 3: Comparison of privacy analyses using Rényi DP (RDP) and Gaussian DP (GDP).

|  | Rényi DP | Gaussian DP | $\delta$ |
|---|---|---|---|
| CIFAR-10 | 7.63 | 7.54 | $10^{-5}$ |
| ECG | 1.76 | 1.62 | $5 \cdot 10^{-4}$ |
| SpeechCommands | 1.47 | 1.39 | $10^{-5}$ |
| FastMRI | 0.67 | 0.16 | $10^{-5}$ |
| PhaseMNIST | 0.53 | 0.51 | $10^{-5}$ |

## A.3 BENCHMARKING COMPLEX-VALUED ACTIVATION FUNCTIONS FOR $\zeta$-DP-SGD

A number of specialised activation functions designed for utilisation with complex-valued neural networks have been proposed in literature. To guide practitioner choice in our newly proposed setting of $\zeta$-DP-SGD training, we here provide activation function benchmarks on the SpeechCommands dataset used in Section 5.3 of the main manuscript. Table 4 summarises these results. We consistently found the inverted Gaussian (iGaussian) activation function to perform best in the $\zeta$-DP-SGD setting. This may be in part due to its bounded magnitude, thereby recapitulating the effect Papernot et al. (2020) discuss for real-valued networks, i.e. that bounded activation functions lead to improved performance in DP-SGD. We leave the further investigation of this finding to future work.

## A.4 BENCHMARKING $\zeta$-DP-SGD ON PHASEMNIST

The MNIST dataset (LeCun et al., 2010) is widely used as a benchmark dataset in real-valued DP-SGD literature. We here therefore show the experimental evaluation of an adapted, complex-valued version of MNIST, which we term PhaseMNIST. The details of how this dataset can be constructed as well as details on the used model architectures are provided in Appendix A.6. In brief, for each example of the original MNIST dataset with label $L_\Re \in \{0,\ldots,9\}$, we obtain the imaginary component by selecting an image with label $L_\Im$ such that $L_\Re + L_\Im = 9$ resulting in an input image arrangement $(0,9),(1,8),\ldots,(9,0)$. Only the label of the real-valued image is used. The results are summarised in Table 5, where we also provide baselines for real-valued MNIST training on the same architecture (with real-valued weights). For simplicity, we have included the RDP and GDP guarantees in the same table.

Table 4: ROC-AUC (mean±STD) of complex-valued activation functions on the SpeechCommand dataset trained with identical settings and the same network architecture over five repetitions with $\zeta$-DP-SGD.

| Activation function | Reference | ROC-AUC |
|---|---|---|
| Separable Sigmoid | Nitta (1997) | $52.9 \pm 0.02\%$ |
| zReLU | Guberman (2016) | $54.7 \pm 0.03\%$ |
| Trainable Cardioid (per-feature bias) | Virtue et al. (2017) | $80.2 \pm 0.02\%$ |
| SigLog | Georgiou & Koutsougeras (1992) | $87.3 \pm 0.01\%$ |
| Trainable ModReLU (per-feature bias) | Arjovsky et al. (2016) | $89.0 \pm 0.01\%$ |
| Cardioid | Virtue et al. (2017) | $89.2 \pm 0.01\%$ |
| Trainable Cardioid (single bias) | Virtue et al. (2017) | $89.4 \pm 0.02\%$ |
| ModReLU (single bias) | Arjovsky et al. (2016) | $89.5 \pm 0.01\%$ |
| cReLU | Trabelsi et al. (2017) | $91.9 \pm 0.01\%$ |
| iGaussian | Virtue et al. (2017) | $\mathbf{93.4 \pm 0.01}\%$ |

Table 5: Results for PhaseMNIST training in a private ($\zeta$-DP-SGD) and non-private (non-DP) fashion. Results for real-valued MNIST are provided for approximate comparison using the same model architecture (but with real-valued weights) trained with identical settings.

| | Accuracy | ROC-AUC | $F_1$-score | Recall | RDP-$\varepsilon$ | GDP-$\varepsilon$ | $\delta$ |
|---|---|---|---|---|---|---|---|
| PhaseMNIST $\zeta$-DP-SGD | $\mathbf{99.0}\%$ | $\mathbf{100}\%$ | $\mathbf{99.0}\%$ | $\mathbf{99.0}\%$ | 0.53 | 0.51 | $10^{-5}$ |
| MNIST DP-SGD | $95.7\%$ | $99.9\%$ | $95.6\%$ | $95.6\%$ | 0.53 | 0.51 | $10^{-5}$ |
| PhaseMNIST non-DP | $\mathbf{99.3}\%$ | $\mathbf{100}\%$ | $\mathbf{99.2}\%$ | $\mathbf{99.2}\%$ | $\infty$ | $\infty$ | 0 |
| MNIST non-DP | $97.4\%$ | $100\%$ | $97.4\%$ | $97.4\%$ | $\infty$ | $\infty$ | 0 |

As with the CIFAR-10 benchmarks in the main manuscript, we found that $\zeta$-DP-SGD outperformed the real-valued network. In this case, additional information is available from the imaginary component of the image in addition to the higher entropic capacity of the network due to the complex-valued weights. A similar phenomenon was observed by Scardapane et al. (2018).

## A.5 WIRTINGER/$\mathbb{CR}$-CALCULUS

In this section, we present key results from Wirtinger (or $\mathbb{CR}$-) calculus which are used in our work. For a detailed treatment, we refer to Kreutz-Delgado (2009).
Consider a function $f : \mathbb{C} \mapsto \mathbb{C}$. As for real-valued functions, the derivative of $f$ at a point $z \in \mathbb{C}$ can be defined as:

$$f'(z) = \lim_{h \to 0} \frac{f(z+h) - f(z)}{h}, h \in \mathbb{C}. \tag{24}$$

If this limit is defined for the (infinitely many) series approaching $z$, $f$ is called *complex differentiable* (equivalently, differentiable *in the complex sense*). If, in addition, $f'(z)$ exists everywhere in the neighbourhood $\mathcal{U}$ of $z$, $f$ is called *holomorphic*. It is also possible to write $z = x + y\,\mathbf{i}$ and to then express $f$ as two real-valued functions $u$ and $v$ of the variables $x$ and $y$:

$$f(x + y\,\mathbf{i}) \coloneqq u(x,y) + v(x,y)\,\mathbf{i}, x, y \in \mathbb{R}. \tag{25}$$

$f'$ can then be written as:

$$\frac{\mathrm{d}f}{\mathrm{d}z} = \frac{\partial u}{\partial x} + \frac{\partial v}{\partial x}\,\mathbf{i}. \tag{26}$$

If this derivative exists at $z$, $f$ is called differentiable *in the real sense*. This interpretation represents $\mathbb{C}$ as $\mathbb{R}^2$ or, more generally, for vector-valued functions, $\mathbb{C}^n$ as $\mathbb{R}^{2n}$. The *Cauchy-Riemann* equations state that, for $f$ to be holomorphic, it must satisfy:

$$\frac{\partial u}{\partial x} = \frac{\partial v}{\partial y} \text{ and } \frac{\partial u}{\partial x} = -\frac{\partial v}{\partial y}. \tag{27}$$

As discussed above, the complex plane does not admit a natural ordering. Hence, the minimisation of a complex-valued function is not defined. Therefore, for complex-valued deep learning, we only consider real-valued (loss-) functions $f : \mathbb{C} \mapsto \mathbb{R}$. By equation (25), $v(x, y)\,\mathbf{i} = 0$. Thus, by the *Cauchy-Riemann* equations, such a real-valued function is only holomorphic if:

$$\frac{\partial u}{\partial x} = \frac{\partial u}{\partial y} = 0. \tag{28}$$

This means that any holomorphic real-valued function must be constant, which invalidates its usefulness for optimisation. The Wirtinger/$\mathbb{CR}$-derivatives provide an alternative interpretation of the *Cauchy-Riemann* equations which allows us to consider holomorphicity and differentiability in the real sense separately. Thus, they recover the usefulness of interpreting $\mathbb{C}^n$ as $\mathbb{R}^{2n}$ while preventing multiplicative penalties on the gradient norm as a consequence of following this interpretation "too closely". We will motivate this somewhat informal notion with an example below. The Wirtinger/$\mathbb{CR}$-derivatives[1] of $f$ are defined as:

$$\frac{\partial}{\partial z} := \frac{1}{2}\left(\frac{\partial}{\partial x} - \frac{\partial}{\partial y}\,\mathbf{i}\right) \text{ and } \frac{\partial}{\partial \overline{z}} := \frac{1}{2}\left(\frac{\partial}{\partial x} + \frac{\partial}{\partial y}\,\mathbf{i}\right). \tag{29}$$

An immediate consequence of this definition is that the *Cauchy-Riemann* equations can be expressed as:

$$\frac{\partial}{\partial \overline{z}} = 0. \tag{30}$$

Therefore, if a function $f$ is holomorphic, $\frac{\partial}{\partial z}$ corresponds to the derivative in the complex sense (that is, $\frac{\mathrm{d}f}{\mathrm{d}z}$) while, if $f$ is differentiable in the real sense, both $\frac{\partial}{\partial z}$ and $\frac{\partial}{\partial \overline{z}}$ are valid (and are conjugates of each other). As stated above, it can be shown that the steepest ascent of $f$ is aligned with $\frac{\partial}{\partial \overline{z}}$. In this sense, $\frac{\partial}{\partial \overline{z}}$ fulfils the role of the $\nabla$ operator for real, scalar-valued loss functions. Evidently, compared to the *actual* gradient of $f$ in the real sense, the following relationship holds:

$$\frac{\partial}{\partial \overline{z}} = \frac{1}{2}\nabla f = \frac{1}{2}\left(\frac{\partial f}{\partial \overline{\theta_1}}, \ldots, \frac{\partial f}{\partial \overline{\theta_n}}\right). \tag{31}$$

However, re-defining $\nabla f := \frac{\partial}{\partial \overline{z}}$ is desirable (and correct, as shown by Brandwood (1983)) . We will motivate this requirement with an example: Let $f_{\mathbb{C}}$ be a function such that:

$$f_{\mathbb{C}}(z) = z\overline{z} = (x + y\,\mathbf{i})(x - y\,\mathbf{i}) = x^2 + y^2. \tag{32}$$

The Wirtinger/$\mathbb{CR}$-derivative of $f_{\mathbb{C}}$ is $\frac{\partial f_{\mathbb{C}}}{\partial \overline{z}} = z$, whose $L_2$-norm is $\|z\|_2 = |z| = \sqrt{|x|^2 + |y|^2}$. The same output can be realised by interpreting $f$ as a function of a real-valued vector $\mathbf{a} = (x, y)^T$:

$$f_{\mathbb{R}}(\mathbf{a}) = \mathbf{a}\mathbf{a}^T = x^2 + y^2. \tag{33}$$

The gradient of $f_{\mathbb{R}}$ is $\nabla f_{\mathbb{R}} = (2x, 2y)$, whose $L_2$-norm is $\|\nabla f_{\mathbb{R}}\| = \sqrt{(2|x|)^2 + (2|y|)^2} = 2\sqrt{|x|^2 + |y|^2} = 2|z|$. This undesirable multiplicative penalty, which would translate to a superfluous multiplicative increase in the noise scale of the GM to preserve DP, is a consequence of "ignoring" the connection between real and imaginary part inherent to complex numbers, but not to components of vectors. In fact, $z\overline{z}$ is *neither* equivalent to $z^2$ (as would be the case if $z \in \mathbb{R}$ where $\forall\, a \in \mathbb{R}, a = \overline{a}$), *nor* is it equivalent to $\langle \mathbf{a}, \mathbf{a}\rangle, \mathbf{a} \in \mathbb{R}^2$, as the complex inner product lacks the bilinearity inherent to its real-valued counterpart. Both complications are avoided by the re-definition of the Wirtinger/$\mathbb{CR}$-derivative as the gradient used for optimisation, which prompts its utilisation in our work.

To exemplify this finding in the training of neural networks, we conducted the following experiment: We generated one thousand points using the function $f(z) = \sin(z) + \mathcal{N}_{\mathbb{C}}(0, 1), z \in \mathbb{C}$. We then trained two separate neural networks to fit the sine function and measured the norms of the gradient vectors after 100 training steps (when gradient norms were empirically highest). One network included complex-valued weights ($\mathbb{C}$), the other simulated complex-valued weights with an additional weight matrix per layer ($\mathbb{R}^{2n}$). Both networks utilised the Cardioid activation function and were

---

[1]The term *derivative* represents an abuse of terminology, as they are formal operators and not derivatives with respect to actual variables. However, the interpretation as derivatives is intuitive, and we will thus retain it.

trained with the Adam optimiser with a learning rate of $0.01$ using a set of identical random seeds over $100$ repetitions for $1000$ steps. We utilised the Mean Squared Error (MSE) loss function given by $f(\mathbf{y}, \hat{\mathbf{y}}) = \frac{1}{2}\|\mathbf{y} - \hat{\mathbf{y}}\|_2, (\mathbf{y}, \hat{\mathbf{y}}) \in \mathbb{C}$. For the $\mathbb{C}$ network, we calculated $\mathbb{CR}$-derivatives, while for the $\mathbb{R}^{2n}$ network, standard gradients were obtained using the PyTorch automatic differentiation system.

Figure 4 shows the gradient norms of the two networks. The $\mathbb{R}^{2n}$-network had significantly higher average gradient norms (Student $t$-test $p = 0.0043$) over the $100$ repetitions.

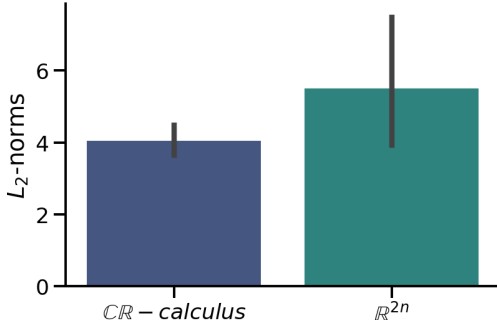

Figure 4: Gradient norms of the $\mathbb{C}$ network calculated using the $\mathbb{CR}$-calculus and of the $\mathbb{R}^{2n}$-network. $N = 100$ repetitions. Significantly higher gradient norms are observed when the $\mathbb{CR}$-calculus is not used (Student $t$-test $p = 0.0043$).

As a note to practitioners, certain deep learning frameworks silently re-scale the Wirtinger/$\mathbb{CR}$-gradient by $2\times$ to avoid user confusion by a lower effective learning rate. To ascertain a correct implementation, we therefore recommend examining this behaviour by testing the gradient norm of known functions.

### A.6 Dataset preparation and Model Training

#### A.6.1 CIFAR-10

**Dataset preparation**  We utilised the CIFAR-10 dataset as described in Krizhevsky et al. (2009). For complex-valued training in $\mathbb{C}$, we either zeroed or copied each image into the imaginary component, while for training in $\mathbb{R}^{2n}$, two separate training channels were used for each tensor. Results in Table 1 are for the zero imaginary/second channel component, but were identical (within $< 1\%$ for all metrics) for the case where the imaginary component was filled with the same image.

**Model training**  For real-valued non-DP training, we used the model described in Papernot et al. (2020) (see Table 2) with the SGD optimiser at a learning rate of $0.1$ with a momentum term of $0.9$ and a cosine learning rate scheduler. The hyperbolic tangent (TanH) activation function was used. For complex-valued non-DP training, we utilised the same architecture and activation functions, albeit with complex weights, or with an additional weight matrix per layer. For DP training, we used an $L_2$ clipping norm of $1.0$, a noise multiplier of $0.61$, a sampling rate of $0.005$ and trained for $20$ epochs. The SGD optimiser at a learning rate of $0.1$ with a momentum term of $0.9$ and a cosine learning rate scheduler was used. Interestingly, this arrangement led to a higher test set performance than reported by Papernot et al. (2020) in the non-DP setting, however we were unable to reproduce the test set accuracy reported in the real-valued DP setting. For training in $\mathbb{R}^{2n}$, we note that multivariate Gaussian noise was used instead of circularly symmetric complex-valued Gaussian noise, and "standard" gradient computations were used instead of Wirtinger derivatives.

#### A.6.2 ECG Dataset

**Dataset preparation**  We utilised the *China Physiological Signal Challenge 2018* (Liu et al., 2018) dataset for this task. We used the *normal* and *left bundle branch block* classes and channel 3. The ECGs were loaded from the provided *Matlab* format using the *SciPy* library and trimmed or padded to a length of $5000$. The *numpy Fast Fourier Transform* implementation was used whereby the

signal was pre-trimmed to length 512. The final dataset consisted of 1012 training examples and 113 testing examples.

**Model training**    We implemented a complex-valued fully-connected neural network architecture consisting of input/hidden layers with $(512, 256, 128)$ ($\zeta$-DP-SGD) units and a single output unit. The cReLU activation function was used both in the non-DP and the $\zeta$-DP-SGD setting. The output layer implemented the magnitude operation followed by a logistic sigmoid activation function. Models were trained using the SGD optimiser at a learning rate of $0.08$ with an $L_2$ regularisation of $5 \times 10^{-3}$ for non-DP training and a learning rate of $0.05$ for $\zeta$-DP-SGD training, respectively. A batch size of $64$ was used for non-private training and a sampling rate of $0.063$ at a noise multiplier of 5 and an $L_2$ clipping norm of $0.5$ for $\zeta$-DP-SGD. $\varepsilon$ was calculated at a $\delta$ of $\frac{1}{1012^{1.1}} \approx 5 \times 10^{-4}$ where 1012 is the number of training samples. Both models were trained for 100 epochs.

### A.6.3    SPEECH COMMAND CLASSIFICATION DATASET

**Dataset preparation**    We used a subset of the *SpeechCommands* dataset (Warden, 2018) as described above, consisting of 2000 samples each from the categories "Yes", "No", "Up", "Down", "Left", "Right", "On", "Off", "Stop", and "Go". Of these, 7200 examples were used as the training test and 800 as the testing set. The waveform data was decoded using the *TensorFlow* library and, where necessary, padded to a length of 16000 samples. The *TensorFlow* implementation of the *Short time Fourier Transform* function was used with a frame length of 255 and a frame step of 128.

**Model training**    For this task, we employed a complex-valued 2D CNN consisting using filters of size $3 \times 3$ without zero-padding and a stride of 1. The convolutional layers had $(8, 16, 32, 64)$ output filters, whereby a MaxPooling layer was used between the second layer and the third layer and an adaptive MaxPooling layer after the final convolutional layer. The convolutional block was followed by a fully connected layer with $64$ units and an output layer of $8$ units. Both employed the iGaussian activation function. The non-DP model was trained at a batch size of $64$ for 10 epochs at a learning rate of $0.1$ using the Stochastic Gradient Descent optimiser, whereas the $\zeta$-DP-SGD network was trained using a sampling rate of $0.009$ for 5 epochs with the same learning rate and optimiser, a noise multiplier of 1 and an $L_2$ clipping norm of 2. We calculated $\varepsilon$ at a $\delta$-value of $10^{-5}$.

### A.6.4    FASTMRI KNEE DATASET

**Dataset preparation**    We utilised the *single coil knee MRI* dataset of the *fastMRI* challenge proposed by Zbontar et al. (2018). We used the reference implementation [2], and employed the default settings using an acceleration rate of $4\times$ and $8\%$ of densely sampled *k*-space center lines in the mask. Masks are sampled pseudo-randomly during training time. The dataset offers $34742$ train and $7135$ validation images.

**Model training**    We changed the *U-Net* network to use complex-valued weights and accept complex-valued inputs instead of the magnitude image employed in the original example. We replaced the original ReLU activation functions with CReLU. In the DP setting, we used a noise multiplier of 1.0, an $L_2$ clipping norm of 1.0 and a sampling rate of $3 \times 10^{-5}$ and calculated the $\varepsilon$ at a $\delta$ of $10^{-5}$. The learning rate was set to $0.001$ using the RMSProp optimiser and a stepwise learning rate scheduler. We trained both in the non-private and the $\zeta$-DP-SGD setting for 30 epochs and disabled the collection of running statistics in the Batch Normalisation layers to render them compatible with DP (in the utilised library, sample sizes of 1 are used, so Batch Normalisation corresponds to Instance Normalisation in this setting).

### A.6.5    PHASEMNIST

**Dataset construction**    As described in Appendix A.4 above, PhaseMNIST is intended as a benchmark dataset for complex-valued computer vision tasks and contains images of handwritten digits from 0 to 9 [3]. The training set consists of $60000$ images and the testing set of $10000$ images. For each

---

[2] https://github.com/facebookresearch/fastMRI/tree/main/fastmri_examples/unet

[3] The version of the dataset used in this study will be made publicly available upon acceptance.

example of the original MNIST dataset, from which PhaseMNIST is constructed, we performed the following procedure: Let $L_\Re \in \{0, \ldots, 9\}$ be the label corresponding to the real-valued image. We then constructed the imaginary component by (deterministically) sampling uniformly with replacement from the set of images whose label $L_\Im$ satisfies $L_\Re + L_\Im = 9$. We used the label of the real-valued image as the label of the overall training example.

**Model training** We used a complex-valued model consisting of three fully connected layers with $(784, 256, 128)$ units and an output layer of $10$ units. The Cardioid activation function was used between layers and the Softmax activation function after the output layer. The model was trained with the Stochastic Gradient Descent optimiser at a learning rate of $0.1$ both for $\zeta$-DP-SGD and for non-private training. The non-private model converged after 3 epochs, whereas the $\zeta$-DP-SGD model required 10 epochs to achieve the same accuracy. The noise multiplier was set to $1.1$ and the $L_2$ clipping norm to $1.0$. The $\varepsilon$ value was calculated at a $\delta = 10^{-5}$. A sampling rate of $0.001$ was used for $\zeta$-DP-SGD, and a batch size of 64 for non-DP training.

### A.7 Software libraries and computational resources used

Implementations of the DP-SGD algorithm, and –by extension– $\zeta$-DP-SGD require access to per-example gradients. We utilised the *deepee* software library (Ziller et al., 2021) to implement $\zeta$-DP-SGD, as it is compatible with arbitrary neural network architectures, including such containing complex-valued weights. We report results using *uniform without replacement* sampling and using the *RDP* option provided by *deepee*. For complex-valued neural network components, the *PyTorch Complex* library (Chatterjee et al., 2021) with *PyTorch 1.9* were used. Standard PyTorch was also used to create the $\mathbb{R}^{2n}$ model architectures for the experiments in Section 5.1. *TensorFlow 2.4* was used for loading data and the *Short Time Fourier Transforms* discussed above, but no neural network components were used from this library. Experiments were carried out in *Python 3.8.5* on a single workstation computer running *Ubuntu Linux 20.04* and equipped with a single *NVidia Quadro RTX 8000* GPU, 12 CPU cores and 64 GB of RAM.

### A.8 Computational considerations

We conclude by presenting a systematic evaluation of the computational considerations incurred by the utilisation of complex-valued neural networks and by the implementation of $\zeta$-DP-SGD using the above-mentioned libraries. Two main sources of computational overhead arise between real-valued and complex-valued neural networks. Complex numbers are internally represented as a pair of 32-bit floating point numbers. This affects inputs and neural network weights. Moreover, even though a complex-valued architecture may contain the same number of parameters as its real-valued counterpart, an increased number of computational operations is required in $\mathbb{C}$. For instance, scalar multiplication requires a single multiplication operation in $\mathbb{R}$. However, in $\mathbb{C}$ it can require up to 4 multiplications (although this can be reduced to 3 multiplications (Higham, 1992)). Performance moreover depends on whether vector hardware is used and whether complex floating point instructions are implemented in the respective framework (e.g., *cuDNN*). Table A.8 shows results for individual matrix multiplication operations and convolutions with real/complex-valued inputs and weight matrices.

Table 6: Average computation times (100 repetitions) for a batched matrix multiplication with batch size 64 and matrix dimensions $(512 \times 512)$ (Linear) and a convolution operation with input dimensions $(64 \times 3 \times 224 \times 224)$ (batch, channel, height, width) and kernel dimensions $(3, 16, 3)$ (in, out, kernel size) (Conv.). $\mathbb{R}$: Real-valued input and weight matrix and $\mathbb{C}$: complex-valued input and weight matrix. Times are given on CPU and GPU.

|  | Linear | | Conv. | |
|  | $\mathbb{R}$ | $\mathbb{C}$ | $\mathbb{R}$ | $\mathbb{C}$ |
| --- | --- | --- | --- | --- |
| CPU | $68.6\ \mu s$ | $1.21$ ms | $31.8$ ms | $214$ ms |
| GPU | $49.6\ \mu s$ | $238\ \mu s$ | $1.38$ ms | $12.5$ ms |

$\zeta$-DP-SGD carries additional overhead as it requires per-sample gradients. In the utilised *deepee* framework, this is realised through dispatching one computation thread per example in the minibatch (more precisely, *lot*) to perform a forward and backward pass, which incurs substantial overhead compared to pure vectorisation. These results are shown in Table A.8. Of note, for the non-private models, the computation time includes the forward pass, backward pass, loss gradient calculation (Mean Squared Error against a vector of dimensions $(64, 1)$) and weight update (Stochastic Gradient Descent). For the $\zeta$-DP-SGD model, the following additional steps occur between the loss gradient calculation and the weight update: gradient clipping, averaging of per-sample gradients, noise application. Moreover, the *deepee* framework requires an additional step between the weight update and the subsequent batch.

Table 7: Average computation times for a model consisting of a 2D convolutional layer with 3 input channels, 32 output channels and a kernel shape of $3 \times 3$ followed by a linear layer with matrix dimensions of $(28800, 1)$ executed on an input of dimensionality $(64, 3, 32, 32)$ (batch, channel, height, width) for 100 repetitions.

|  | Non-DP | | ($\zeta$-)DP-SGD | |
|  | $\mathbb{R}$ | $\mathbb{C}$ | $\mathbb{R}$ | $\mathbb{C}$ |
| --- | --- | --- | --- | --- |
| CPU | 156 ms | 1.45 s | 27.6 s | 1.08 min |
| GPU | 187 ms | 588 ms | 8.52 s | 17.1 s |

