# OpenReview forum: "Complex-valued deep learning with differential privacy"
_ICLR.cc/2022/Conference — ICLR 2022 Submitted_

### Official Review · Reviewer_SGmD · 2021-10-24

**Correctness:** 4
**Technical Novelty And Significance:** 4
**Empirical Novelty And Significance:** 4
**Recommendation:** 8
**Confidence:** 4

**Main Review:**

Overall, the paper is very well-written and insightful. The contents are complete, covering all important aspects of DP deep learning. Especially the complex-valued Gaussian distribution, the choice of L2 over L1 sensitivity, the seamless connection between complex-valued privacy accountant and existing ones, the choice of Wirtinger calculus, etc. are very clear and exciting.

The definition of $\xi$-DP is exactly $(\epsilon,\delta)$-DP but extended to the complex domain. So this is moderately interesting. The true highlights of this paper are the new Gaussian mechanism (as well as the new DP optimizers) and good motivation of complex-valued deep learning.

As for the weakness, I think there are 2 points:
(1) this paper under-sells its strength in generalizability. Although the author discussed other privacy accounting like f-DP, I believe more other accounting like Gaussian DP, Fourier accountant and others should be mentioned. Another key part is the optimizer. If there is no fundamental difficulty, the authors should discuss $\xi$-DP-Adam/Momentum/Adagrad and new clipping method like global clipping. If there is a challenge to design new complex-valued DP optimizers, the authors should be state clearly the reasons.
(2) the experiments are exciting but can be improved. First of all, MNIST is too simple even for DP learning. I would expect to understand the performance on CIFAR10. Notice that while DP learning can get 99% on MNIST, it at most gets 70% on CIFAR10 without transfer learning. Also Table 2,3,5 look incomplete to me. Why MNIST has accuracy but these tables have ROC? Please include accuracy and possibly other metrics in all tables.

**Summary Of The Paper:**

This paper proposes a novel and comprehensive direction in private deep learning, namely the complex-valued private learning. A new notion of privacy--$\xi$-DP, is defined and carefully analyzed. For this new privacy, a new Gaussian mechanism allows novel DP optimizers to be developed. Extensive experiments demonstrate good performance by combining differential privacy and complex-valued deep learning.

**Summary Of The Review:**

The paper is very well-written and insightful. It provides new DP notion, new Gaussian mechanism, new DP optimizer and exciting experiments. Minor weakness exists: the extensions are not well-discussed and experiments can be enhanced.

---

> ### Author Response · Authors · 2021-11-18
> **Response to Reviewer SGmD**
>
> We warmly thank the reviewer for their kind remarks about our work and the suggestions.
>
> > The definition of ξ-DP is exactly (ϵ,δ)-DP but extended to the complex domain. So this is moderately interesting. The true highlights of this paper are the new Gaussian mechanism (as well as the new DP optimizers) and good motivation of complex-valued deep learning.
> >
>
> Regarding this point, we have improved the manuscript to highlight that the extension to the complex domain is natural, yet provides some additional benefits: the utilisation of complex-valued noise (instead of multivariate real-valued noise) offers improved utility for the same privacy guarantees by leveraging the properties of the Hermitian inner product and the "connection" of the real and imaginary components of complex numbers. Please also compare our responses to reviewers n9FY and kQQG.
>
> > As for the weakness, I think there are 2 points: (1) this paper under-sells its strength in generalizability. Although the author discussed other privacy accounting like f-DP, I believe more other accounting like Gaussian DP, Fourier accountant and others should be mentioned. Another key part is the optimizer. If there is no fundamental difficulty, the authors should discuss ξ-DP-Adam/Momentum/Adagrad and new clipping method like global clipping. If there is a challenge to design new complex-valued DP optimizers, the authors should be state clearly the reasons.
> >
>
> We thank the reviewer for the input on this point and agree that these are important discussion points. On the topic of Gaussian DP, we have included an additional theorem (Theorem 3) for Gaussian DP, showing how GDP enables a tighter privacy analysis of the complex Gaussian mechanism. This was also requested by reviewer n9FY.  Moreover, we now explicitly discuss that our technique is compatible with alternative optimisers and clipping methods, as well as the Fourier accountant in Section 4.1..
>
> > (2) the experiments are exciting but can be improved. First of all, MNIST is too simple even for DP learning. I would expect to understand the performance on CIFAR10. Notice that while DP learning can get 99% on MNIST, it at most gets 70% on CIFAR10 without transfer learning.
>
> We thank the reviewer for this suggestion. We replaced the section on MNIST (which we have moved to the Appendix) with benchmarks on CIFAR10 (Section 5.1). We found that training complex-valued neural networks with $\zeta$-DP-SGD leads to superior test set performance compared to training real-valued neural networks with DP, even when the imaginary component of the input images was zero. This result confirms our theoretical analysis and highlights that our proposed technique can be used to obtain superior model utility for the same privacy guarantees, even for real-valued tasks.
>
> > Also Table 2,3,5 look incomplete to me. Why MNIST has accuracy but these tables have ROC? Please include accuracy and possibly other metrics in all tables.
>
> We welcome the reviewer's proposal on this point. For completeness' sake, we have now included Accuracy, Recall, $F_1$-score and ROC-AUC as well as the results from the improved Gaussian DP accounting to Table 1, which now incorporates all results from the classification experiments.

---

### Official Review · Reviewer_kQQG · 2021-10-25

**Correctness:** 4
**Technical Novelty And Significance:** 2
**Empirical Novelty And Significance:** 4
**Recommendation:** 6
**Confidence:** 3

**Main Review:**

I found the paper to be well written and the problem of  "complex DP" well-motivated.
The idea of implementing DP-SGD with the complex Wirtinger notion of derivative is particularly elegant and appealing.

However, I would like to contend that there the introduced ideas $\zeta$-DP and complex Gaussian mechanism are not inherently complex.
Indeed, by considering the identification $\mathbb{C}^n = \mathbb{R}^{2n}$. The complex Gaussian mechanism reduces to the standard Gaussian mechanism when the dimension is doubled. The same is also true for $\zeta$-DP and 'vanilla' DP in real spaces. This is reflected, for example, in the proof of Theorem 1, which is essentially identical to the proof in Balle & Wang (2018) (except the dimension has doubled).
I would be very happy to be proven wrong on this point and I expect the authors' response.

In light of the above though, it seems that the main contribution of the paper is to suggest an alternative to DP-SGD, at least when the dimension is even. Since the functions to optimize are necessarily real-valued, otherwise, it's not even clear what is being optimized, the authors suggest using the conjugate of the complex derivative. For real-valued functions, the conjugate is aligned with the regular gradient. Again, I don't find this setting particularly 'complex-valued'. Other than making the gradient smaller by a constant what are the benefits of using the Wirtinger derivative?
I would at least expect to see some experiments which show that $\zeta$-DP-SGD on $\mathbb{C}^n$ outperforms 'vanilla' DP-SGD on $\mathbb{R}^{2n}$, when optimizing the same function, with the same privacy budget.
Currently, the experiments only demonstrate that $\zeta$-DP-SGD well on a variety of tasks. As the authors note this can be due to the extra information contained in the additional dimensions.

One final point, which is a matter of personal taste. The title of the paper "COMPLEX-VALUED  DEEP  LEARNING..." is not appropriate, and even misleading, in my eyes. Nothing in the suggested framework is specific to deep learning. While the authors do use neural networks (none of which are particularly deep) in their experiments, it is not the main focus of the paper.





**Summary Of The Paper:**

The paper proposes a new framework, which extends differential privacy to complex-valued functions. The authors name this framework $\zeta$-DP and introduce their main privacy mechanism, the complex Gaussian mechanism. The authors also show how to adapt the private gradient descent algorithm into a private algorithm to minimize real-valued functions, defined on the complex plane.
It is then shown, experimentally, that the new algorithms and notions perform very well on a variety of tasks related to signal processing (and hence to complex-valued functions).


**Summary Of The Review:**

The authors suggest a new framework for differential privacy in complex spaces, which works well with complex notions of derivatives.
However, as far as I have gathered, this is not particularly different than the existing notion of differential privacy in real spaces and I was not convinced there is an actual benefit to using the complex derivatives.

---

> ### Author Response · Authors · 2021-11-18
> **Response to Reviewer kQQG, Part 1.**
>
> We would like to warmly thank the reviewer for their constructive remarks and suggestions on our manuscript, and respond in detail to the concerns raised.
>
> > However, I would like to contend that there the introduced ideas ζ-DP and complex Gaussian mechanism are not inherently complex. Indeed, by considering the identification Cn=R2n. The complex Gaussian mechanism reduces to the standard Gaussian mechanism when the dimension is doubled. The same is also true for ζ-DP and 'vanilla' DP in real spaces. This is reflected, for example, in the proof of Theorem 1, which is essentially identical to the proof in Balle & Wang (2018) (except the dimension has doubled). I would be very happy to be proven wrong on this point and I expect the authors' response.
> > In light of the above though, it seems that the main contribution of the paper is to suggest an alternative to DP-SGD, at least when the dimension is even. Since the functions to optimize are necessarily real-valued, otherwise, it's not even clear what is being optimized, the authors suggest using the conjugate of the complex derivative. For real-valued functions, the conjugate is aligned with the regular gradient. Again, I don't find this setting particularly 'complex-valued'. Other than making the gradient smaller by a constant what are the benefits of using the Wirtinger derivative? I would at least expect to see some experiments which show that ζ-DP-SGD on Cn outperforms 'vanilla' DP-SGD on R2n, when optimizing the same function, with the same privacy budget. Currently, the experiments only demonstrate that ζ-DP-SGD well on a variety of tasks. As the authors note this can be due to the extra information contained in the additional dimensions.
>
> The identity $\mathbb{C}^n=\mathbb{R}^{2n}$ is indeed commonly used, and employing two-dimensional (that is, 2-channel) real-valued weight matrices to represent complex-valued weights in neural networks instead of complex-valued weight matrices is common practice. However, there are fundamental differences between the two representations.
> In $\mathbb{R}^{2}$, matrix multiplication has four degrees of freedom (scaling, reflection, rotation and shear). Complex multiplication only has two degrees of freedom (scaling and rotation). This is a consequence of the fact that the inner product in $\mathbb{R}$ is symmetric and bilinear, whereas in $\mathbb{C}$ the inner product is *conjugate* symmetric and not bilinear.  The resulting reduction in degrees of freedom allows us to design the complex Gaussian Mechanism, which adds noise scaled by $\sqrt{2}$ to each component a complex number, instead of necessitating adding noise scaled by $2$ to each component of a vector, which would be required in the case we actually used the identity $\mathbb{C}^n=\mathbb{R}^{2n}$ without utilising Hermitian inner products. Therefore, the complex Gaussian Mechanism is not "just" the Gaussian Mechanism in two dimensions (which would be a spherical real-valued multivariate Gaussian, not a circular complex-valued Gaussian). We have clarified this in Definition 7 of the manuscript.
> The utilisation of Wirtinger derivatives stems from the same concern: If the structure of $\mathbb{C}$ is not taken into consideration, the derivative or gradient magnitude may be miscalculated for some operations, but not others. This phenomenon, observed even in works which correctly model complex-valued operations as operations in $\mathbb{R}^{2n}$, may not be problematic if the only consequence is, for example, a change in the effective learning rate. In privacy-relevant scenarios however, we contend that there should be "no room for error".  We thus decided to formalise the utilisation of the Wirtinger calculus, which guarantees that sensitivity calculations will always be correct. We have augmented the discussion of the example in Appendix A.5 to clarify this fact. Moreover, we have included a new Figure (Figure 4, Appendix A.5) which experimentally demonstrates that -in practice- the utilisation of $\mathbb{R}^{2n}$ expectedly leads to higher gradient norms than the Wirtinger derivative.
> Regarding the reviewer's suggestion to experimentally demonstrate that the usage of the complex Gaussian mechanism combined with the increased learning capacity of neural networks leads to improved performance, we benchmarked the performance of real-valued vs. complex-valued neural networks trained with DP, including a comparison to a network utilising multivariate real-valued Gaussian noise and "standard" gradients in $\mathbb{R}^{2n}$. We found that the complex-valued neural network trained with $\zeta$-DP-SGD outperformed both the real-valued DP network and the network "simulating" $\mathbb{C}$ as $\mathbb{R}^{2n}$, even on real-valued inputs. These results, indicating that our proposed technique represents an optimal trade-off between the learning capacity of complex-valued neural networks and the noise required for DP, are presented in Section 5.1. of the main manuscript.

---

> > ### Author Response · Authors · 2021-11-18
> > **Response to Reviewer kQQG, Part 2.**
> >
> > (continued from Part 1 above as a response to the Reviewer's first point)
> >
> > In addition, we would like to clarify the point about the real-valued loss function raised by the reviewer. The loss function itself is not real-valued, in fact all its arguments are complex-valued. Only its output is real-valued. Therefore, the gradient with respect to the network weights is a complex-valued vector. The conjugate gradient is -in this case- not identical to a real-valued loss function case (where conjugate and regular gradient are indeed aligned). Thus, $\zeta$-DP-SGD is not identical to real-valued DP-SGD in two channels. We have clarified this fact in Section 4.2..
> >
> > Our last comment on the non-equivalence of $\mathbb{C}^n$ and $\mathbb{R}^{2n}$ concerns the fact that real-valued activation functions are not seamlessly transferrable to the complex domain. For instance, the ReLU activation function applied independently to two channels is not identical to a "complex ReLU", as the $\max$-function is not defined over the complex numbers. This leads to phase cancellation and thus information loss. Instead, only the correct utilisation of "actual" complex numbers and corresponding activations (in the case of ReLU, the directly corresponding, phase-preserving activation function is the Cardioid) can fully leverage the peculiarities of the complex domain.
> >
> > > One final point, which is a matter of personal taste. The title of the paper "COMPLEX-VALUED DEEP LEARNING..." is not appropriate, and even misleading, in my eyes. Nothing in the suggested framework is specific to deep learning. While the authors do use neural networks (none of which are particularly deep) in their experiments, it is not the main focus of the paper.
> > >
> >
> > We appreciate the reviewer's input on this point. We would like to propose changing the title to:
> >
> > **$\zeta$-DP: Complex-Valued Differential Privacy and its Applications to Neural Network Training**

---

> > > ### Comment · Reviewer_kQQG · 2021-11-23
> > > **Response to Rebuttal**
> > >
> > > Thank you for the thorough and illuminating response.
> > > The added details and the changes to the manuscript have clarified many things.
> > >
> > > I agree with the points you raise in your response. Specifically, that there is some inherent difference between the real and complex case. Though, as far as I can see it, this difference ends up manifesting in constant factors, both in the privacy budget and in the learning rate of DP-SGD.
> > >
> > > Having said that, the experiments do show that this constant gain can be translated into an actual, non-negligible, gain in performance.
> > > I thus prefer to err on the authors' side and have decided to raise my score accordingly.

---

### Official Review · Reviewer_n9FY · 2021-11-02

**Correctness:** 4
**Technical Novelty And Significance:** 2
**Empirical Novelty And Significance:** 2
**Recommendation:** 5
**Confidence:** 4

**Main Review:**

Pros: This paper is well structured and its motivation is clear. The experimental results show that the proposed algorithm is possible to achieve high utility under tight privacy guarantees.

Cons: More significant results are needed. My main concerns are as follows:

- On the theory side, the idea seems too natural. It would be better if authors can provide a tighter bound for the privacy loss.
- On the experiment side, to make the results more convincing, please provide more comprehensive experiment results, such as accuracy for differen

**Summary Of The Paper:**

This paper proposes a complex-valued DP-SGD mechanism, named \zeta-DP-SGD, to train privacy-preserved neural networks on complex-valued data.  In particular, the authors introduce a complex Gaussian mechanism with DP and Renyi-DP properties, \zeta-DP, to extend DP to the complex domain. \zeta-DP allows the re-use of prior theoretical results and software implementations.

**Summary Of The Review:**

Although the privacy-preserved complex-valued DL is an interesting problem, I’m not sure this paper reports enough contribution for pushing the development of this research field. The theory for complex Gaussian Mechanism seems to re-use the prior theoretical results. The used gradient computation method based on Wirtinger calculus is similar to the previous works on complex-valued deep learning.

---

> ### Author Response · Authors · 2021-11-18
> **Response to Reviewer n9FY**
>
> We would like to thank the reviewer for their input on our manuscript, and address their remarks below:
>
> > On the theory side, the idea seems too natural. It would be better if authors can provide a tighter bound for the privacy loss.
>
> We thank the reviewer for this suggestion. Indeed, the complex Gaussian mechanism naturally extends its real-valued counterpart. To address the reviewer's suggestion regarding providing a tighter bound on the privacy loss, we have made the following modifications:
>
> 1. We have added an additional Theorem (Theorem 3) in which we analyse the complex Gaussian Mechanism using Gaussian DP, which offers the tightest possible privacy loss bound. To experimentally demonstrate the benefits of this new analysis, we have repeated all experiments using Gaussian DP accounting and show that this accounting technique indeed leads to lower $\varepsilon$-budget consumption. These results can be found in Appendix A.2, Table 3.
> 2. We have clarified that the main benefit of using complex-valued neural networks and the complex Gaussian Mechanism/$\zeta$-DP(-SGD) is that neural networks with complex-valued weights are more parameter-efficient by having an additional degree of freedom (the ability to represent rotation and not just scaling) per weight compared to scalar, real-valued weights. Our theoretical contribution of the complex Gaussian Mechanism allows one to obtain the same privacy guarantees as real-valued neural networks and, at the same time, a superior learning capacity and thus, increased utility for the same privacy budget. These facts are discussed in Definition 7 and Section 5.1., as well as in Appendix A.5.
>
> > On the experiment side, to make the results more convincing, please provide more comprehensive experiment results, such as accuracy for differen
> >
>
> Unfortunately, this remark by the reviewer is not complete. We assume that the reviewer is requesting additional metrics to be included, a suggestion which was also remarked by reviewer SGmD. We have now included Accuracy, $F_1$-score, ROC-AUC and Recall scores for all classification experiments (see Table 1 of the main manuscript). Additionally, we augmented the benchmarks by including experiments on the CIFAR-10 dataset, as requested by reviewer SGmD. We found that complex-valued neural networks trained with $\zeta$-DP outperform real-valued neural networks trained with DP by 3% in terms of accuracy for the same privacy budget. This finding underscores that $\zeta$-DP enables training models with improved utility for the same privacy budget.
>
> > Although the privacy-preserved complex-valued DL is an interesting problem, I’m not sure this paper reports enough contribution for pushing the development of this research field. The theory for complex Gaussian Mechanism seems to re-use the prior theoretical results. The used gradient computation method based on Wirtinger calculus is similar to the previous works on complex-valued deep learning.
> >
>
> We respectfully disagree with this point of view. To analyse the novel privacy mechanism we introduce, it is necessary to employ established DP theory. However, no prior work has formally introduced the concepts of the complex Gaussian mechanism and $\zeta$-DP-SGD or shown detailed experiments on a broad variety of complex-valued neural network applications trained under DP guarantees. Moreover, the utilisation of Wirtinger calculus for complex-valued deep learning may have been previously described, but the theoretical analysis of its positive effect on the sensitivity of the neural network which promotes utility has so far not been leveraged for DP (compare also the supplementary discussion of this topic in Appendix A.5). Last but not least, no previous works have suggested utilising the combination of the increased expressivity of complex-valued neural networks with the complex Gaussian mechanism to obtain improved model accuracy and utility for the same privacy budget. The complex Gaussian mechanism is not equivalent to the Gaussian mechanism in two dimensions. Our theoretical analysis permits us to scale the noise added to each component of the complex-valued gradient vector by $\sqrt{2}$ instead of having to scale the noise of a gradient vector in $\mathbb{R}^{2n}$ by $2$.

---

### Author Response · Authors · 2021-11-18
**Thank you to reviewers and summary of changes**

We would like to warmly thank the reviewers and area chairs for their constructive criticism of our work. Due to the inclusion of additional theoretical results, experiments and discussion points, we slightly restructured the paper as follows:

- We shortened the theory section due to the inclusion of an additional theorem. All theorems now have proof sketches, with the full proofs given in the Appendix.
- We replaced the section on PhaseMNIST with benchmarks on CIFAR-10 as requested by reviewer SGmD. The PhaseMNIST results have been moved to the Appendix.
- We consolidated the results from the classification experiments (Sections 5.1-5.3) into one Table (Table 1) and included additional metrics, as requested by reviewers n9FY and SGmD.
- As proposed by Reviewer kQQG, we have modified the title of the manuscript in the submitted PDF version. This is not (yet) mirrored in the OpenReview submission system.
- Similarly, we have corrected the abstract to include the new findings. This is also not (yet) visible in OpenReview and we refer to the PDF.

We address all reviewer remarks in detail below. Please do not hesitate to contact us with any additional remarks or questions.

With kind regards and many thanks in advance,

The Authors.

---

### Decision · Program_Chairs · 2022-01-20

**Decision:**

Reject

**Comment:**

The reviewers in general agree that the proposed complex valued DP method is interesting and novel. However, there are two key concerns due to which the paper might not be ready for publication at ICLR: a. the key technical contribution of the work is not clear, as the methods seem relatively straightforward extension of real valued DP methods to complex valued domains.
b. More importantly, the experimental results (and hence the motivating applications) are not convincing and do not strongly support the claims of i) complex data provides more flexibility and hence provide better model, ii) proposed method is accurate.
For example, the accuracy numbers for SpeechCommands even without DP seem quite low. For example, standard methods like matchboxnet for keyword detection have accuracy numbers in the range of 97%. While the work considers a subset of keywords, but it would be important to show how the standard methods work on this dataset. If the gap is this large, then the case for using complex valued datasets itself is weak.
Similarly, on CIFAR10 it seems like that the considered architecture is quite poor as the accuracy is just ~80% while most standard architectures get >93% on the dataset. So the experiment claims of the paper might not hold for practically relevant architectures.